# A Deep Learning Framework to Remove the Off-Focused Voxels from the 3D Photons Starved Depth Images

Suchit Patel [1,2,†] , Vineela Chandra Dodda [1,†] , John T. Sheridan [3] and Inbarasan Muniraj [1,4,*]

[1]    Department of Electronics and Communication Engineering, School of Engineering and Science,
       SRM University AP, Amaravathi 522240, India; suchit.patel@poornima.org (S.P.);
       vineelachandra_dodda@srmap.edu.in (V.C.D.)
[2]    Department of Computer Engineering, Poornima College of Engineering, Jaipur 302022, India
[3]    School of Electrical and Electronic Engineering, College of Architecture and Engineering,
       University College Dublin, D4 Belfield, Ireland
[4]    LiFE Laboratory, Department of Electronics and Communication Engineering, Alliance College of
       Engineering and Design, Alliance University, Bengaluru 562106, India
[*]    Correspondence: inbarasan.muniraj@alliance.edu.in
[†]    These authors contributed equally to this work.

**Abstract:** Photons Counted Integral Imaging (PCII) reconstructs 3D scenes with both focused and off-focused voxels. The off-focused portions do not contain or convey any visually valuable information and are therefore redundant. In this work, for the first time, we developed a six-ensembled Deep Neural Network (DNN) to identify and remove the off-focused voxels from both the conventional computational integral imaging and PCII techniques. As a preprocessing step, we used the standard Otsu thresholding technique to remove the obvious and unwanted background. We then used the preprocessed data to train the proposed six ensembled DNNs. The results demonstrate that the proposed methodology can efficiently discard the off-focused points and reconstruct a focused-only 3D scene with an accuracy of 98.57%.

**Keywords:** photons counting imaging; deep learning; off-focused removal; dense neural network; 3D reconstruction

## 1. Introduction

Integral Imaging (II) is an optoelectronic-based three-dimensional (3D) imaging technique that captures a 3D object and reconstructs several 3D sectional (or depth) images at good resolution, in real time. As a first step, this method records two-dimensional (2D) images, often known as Elemental Images (EIs), either by employing a lenslet array (for the single shot) or by mechanically moving the camera (for multiple shots) [1–4]. The corresponding single-shot approach is relatively faster but often suffers from poor spatial resolution. To alleviate this, several studies have been proposed to enhance the reconstructed image quality [5,6]. Alternatively, multiple shot imaging, also known as the computational integral imaging (CII) method, provides good resolution but increases the computational burden, as multiple high-quality two-dimensional (2D) images have to be captured and processed [7,8]. Nevertheless, owing to the simplified nature of the image capturing and reconstruction processes, II has gained wide attention among researchers from several scientific areas such as biomedical imaging, remote sensing, autonomous driving, 3D displays, and televisions, to name a few [4].

For several biomedical applications, imaging the samples with an external higher-intensity light source is not optimal, as the light intensity may damage the tissues. In such applications, imaging and reconstructing 3D scenes using lower-intensity light becomes necessary [9]. Several studies have been conducted to demonstrate the feasibility of single or few photons imaging experimentally [10] and computationally [11]. Such approaches have also been combined with CII for 3D imaging, which is known as Photons Counted

Integral Imaging (PCII) [12]. Thereafter, several studies have been performed using PCII for various imaging applications [13,14] as it was shown that such a system reconstructs 3D images even with a low number of photon counts [15].

In principle, CII-based reconstructed 3D depth or 3D sectional images contain both the focused and off-focused (or out-of-focus) pixels, simultaneously. Off-focused pixels often look blurred and therefore do not convey acceptable information about the scene. Few studies have been carried out to efficiently remove the off-focused points from reconstructed 3D images. For instance, Faliu Yi et al. proposed a simple but efficient approach to reconstructing (grayscale) depth images without the off-focused points [16]. Previously, we also demonstrated a subpixel-level three-steps-based statistical approach to efficiently remove the off-focused points from the 3D sectional images in color (RGB) format [17]. We note that both of these previous approaches are subjective, as they involve performing manual calculations of algorithmic parameters such as mean, variance, threshold, etc., which is time-consuming and also varies according to the scene [17].

Intuitively, when the complexity of a problem increases, the time and space complexity required to solve the problem also increase. Mathematical modeling of such problems can be tiresome, demanding more manual inputs. Recently, with advancements in information technology, a new era of automation is growing exponentially. The automation process for any problem varies from simple logic modeling to complex deep learning networks [18]. Such approaches have also been investigated by optical imaging scientists for various image-based applications. For instance, in [19], a DL model was developed to enhance the resolution of an integral imaging-based microscopic system. In [20], the authors demonstrated that DL algorithms can be used for automatic object detection and segmentation. Further, in [21], DL was applied to detect and classify the objects from degraded environments such as low-light illumination and in the presence of occlusions. In [22], for the first time, we developed a DL framework for denoising the computational 3D sectional images. Inspired by these studies, in this work, we developed a novel deep learning framework to efficiently remove the off-focused portions (pixels) from the reconstructed 3D sectional images.

## 2. Methodology

### 2.1. Photon Counted Integral Imaging

Integral Imaging can be realized in two ways, i.e., either a single-shot approach using a lenslet array or a multiple shots approach in which the object is scanned using an imaging sensor or a commercial camera. In this work, we have used the latter approach to achieve higher spatial resolution [8,12]. This approach requires a camera that translates in both the horizontal and vertical directions to capture the multiple 2D images of a 3D scene; see Figure 1. The recorded images are often known as elemental images [EIs] or as an elemental image array [EIA]. These EIs are then used to reconstruct a 3D sectional image. For a 3D scene reconstruction, several techniques have been proposed in the literature [3]. As previously mentioned, in this work, we also have used the photon detection statistical approach (as described in [9,11]) to reconstruct a 3D scene that resembles a scene from ultralow-light illumination conditions. It is known that the arrival of the photon to the imaging sensor is purely a random process, and therefore photons counted images can be modeled using the Poisson distribution (PD) [11]. Let the total number of photons captured in an image be denoted as $n_p$. The probability of counting photons at any arbitrary pixel location (i.e., $f(x,y)$) is defined as follows:

$$Poisson(\lambda(x,y) = EI(x,y) \times n_p) = \frac{[\lambda]^{f(x,y)} \times e^{-\lambda(x,y)}}{f(x,y)!} \tag{1}$$

where $\lambda$ denotes the Poisson parameter at any given arbitrary pixel location, which can be computed by multiplying a normalized input image (in our case, EI) and the expected number of photons per scene [11]. Once the photon counting is applied to the captured

EIs, we then use the maximum likelihood estimation (MLE) technique to reconstruct the photon-counted 3D sectional images. Mathematically, this process is described using [23]:

$$MLE\{I_P^Z\} = \frac{1}{n_p VT} \sum_{v=1}^{V} \sum_{t=1}^{T} C_{vt}(x + v(\frac{s_x}{MF}), y + t(\frac{s_y}{MF})) \tag{2a}$$

where $VT(x,y)$ represents the number of overlapped values in each pixel of the reconstructed sectional image. Subscripts $v$, $t$ denote the location of EI in the pickup grid and MF represents the magnification factor. The shift positions are denoted as

$$s_x = \frac{p_x \times f}{p_s \times d}, s_y = \frac{p_y \times f}{p_s \times d} \tag{2b}$$

where $p_x$, $p_y$ represents the distance between two consecutive image sensor positions. Notably, $p_s$, $f$, and $d$ denote the pixel size of an image sensor, the focal length of the lens, and the distance between the pick-up grid and the image plane (see Figure 1), respectively [24]. Meanwhile, $C_{vt}(.)$ is the photon counted pixel value in the $vt^{th}$ elemental image. A detailed photon counting 3D integral imaging and reconstruction is presented in [11,23], and therefore is not discussed in detail here.

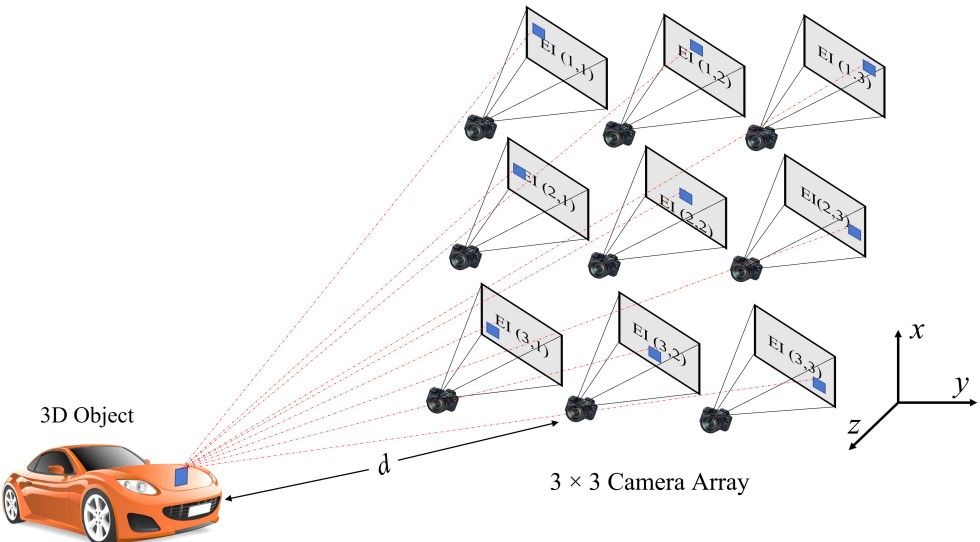

**Figure 1.** Computational Integral Imaging setup ($3 \times 3$ camera array).

### 2.2. Deep Neural Network

In this section, we describe the opted Dense Neural Network (DNN) for removing the off-focused points from the 3D sectional images. In principle, DNN mimics the human brain, which consists of several neurons (which are self-optimized through the learning process), thus providing better accuracy and precision. It is known that DNN is a type of Artificial Neural Network (ANN) that has more than one hidden layer (HL) between input and output [25]. Each HL can have $n$ number of neurons (i.e., dense units) which are connected to every other neuron in the adjacent layer. This formation resembles a web-like structure (see Figure 2) that helps DNN to implement logical operations to establish a non-linear relationship between inputs and outputs. Further, it is known that each neural unit performs a matrix–vector multiplication with an output of the previous layer, and this matrix is updated with each iteration (or epoch) using a backpropagation process. The backpropagation process computes the gradient of the loss function with respect to the single input and output. It is known that, based on our requirements, multiple hyperparameters can also be opted in the dense layers, such as the no. of neural units, the

activation function, the kernel initializer, etc. [26]. Mathematically, DNN is defined by the prediction equation as follows:

$$Y(I_i) = F_n(\dots F_2(W_{(2)}F_1(W_{(1)}[I_i] + b_1) + b_2)\dots) \tag{3}$$

where $Y(I_i)$ is the final prediction (output), $F_n$ is a function that defines output in terms of weights and bias. $W_n$ and $b_n$ denote the $n^{th}$ layer's weight and bias, respectively. $I_i$ represents the input sectional images.

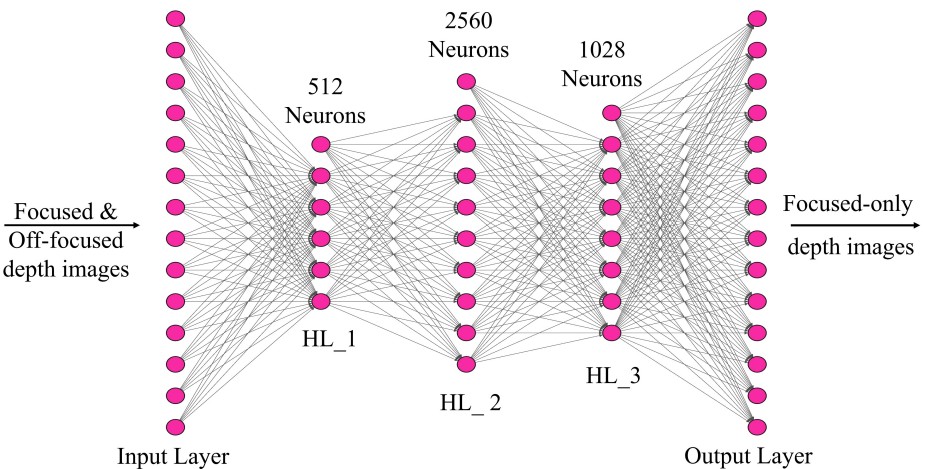

**Figure 2.** Proposed DNN Architecture. HL denotes the hidden layers.

Figure 3 depicts the flowchart of our proposed work. Notably, the proposed ensembled deep neural network is trained (in a supervised manner) using the conventional 3D sectional images from various depth locations and the corresponding focused images (labels).

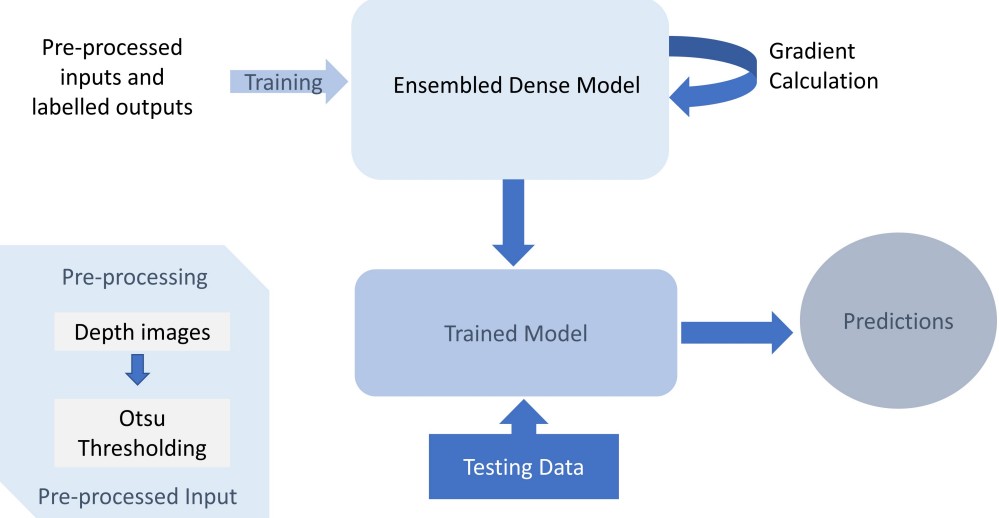

**Figure 3.** Flowchart of our proposed work.

It is known that data pre-processing enhances the accuracy of the network; therefore, we used the Otsu thresholding algorithm [27] to remove the unwanted (obvious) background from the 3D sectional images. In this work, we employed an ensembled DNN model that comprises six different DNN models, each trained with its own set of training datasets. It is known that, in the training process, the selection of cost function is of paramount importance to obtain optimum weights and bias [22]. In the literature, several optimization algorithms were proposed to minimize the cost function, such as the

gradient descent, stochastic gradient descent, Adaptive Gradient Algorithm (ADAGRAD), and Adaptive Moment Estimation (ADAM), to name a few [28–30]. In this work, we opted for an ADAM optimizer to update the weights and bias [22], and a standard Mean Squared Error (MSE) was used as the cost function in our training process.

## 3. Experimental Results

The 3D scene used in our experiment contains two toy cars and one toy helicopter [24]. These objects were placed at different distances, such as 280, 360, and 430 mm from the sensor. We note the imaging sensor size is 22.7 × 15.6 mm and the effective focal length is 20 mm. The pitch (i.e., the distance between two consecutive sensor's positions) is 5 mm. The results are obtained by performing simulations on an Intel® 216 CPU @2.10 GHz (2 processors) with 256 GB RAM. The software used is Spyder integrated development environment from Anaconda Navigator.

As previously mentioned, the proposed ensembled DNN model consists of six different DNN models that have the same architecture and hyper-parameter configuration, which were tuned using the Bayesian Optimization (BO) tuner [31]. It is known that the BO uses the standard Gaussian process to tune the hyper-parameters. Instead of selecting random combinations of hyper-parameters, such as Random Search or Hyperband, BO initially selects the random combination of hyper-parameters. The future combinations are selected based on their performance, such that either the optimal hyper-parameters or the maximum allowed trials are reached. The estimated optimal values for BO from our simulations are given in Table 1. We note the individual DNN model in the Ensembled Super-DNN model was developed using the optimal values from the BO. The individual DNN model summary is given in Table 2. In our simulations, we achieved the off-focused removal accuracy of 98.57% for CII sectional images and 98% for the sectional images based on PCII. We also estimated the computational complexity of our proposed method. Our model consumes 2 s per epoch, resulting in a total computational time of 200 s (for training), and the testing is completed in less than a second. Furthermore, we note that the computational time can be optimized by altering the system configurations.

**Table 1.** Optimal Hyper-parameters.

| Hyper-Parameter | Optimised Value from BO Tuner |
| --- | --- |
| Units of hidden layer | 512, 2560 and 1028 |
| Activation Function (hidden layer, output) | ReLU, Linear |
| Learning rate | $1 \times 10^{-2}$ |
| Optimizer | Adam |
| No. of epochs | 100 |
| Batch size | 1 |

**Table 2.** Individual DNN model summary.

| Model: "Sequential" | | |
| --- | --- | --- |
| Layer (Type) | Output Shape | Param # |
| dense (Dense) | (None, 512) | 36032000 |
| dense_1 (Dense) | (None, 2560) | 1313280 |
| dense_2 (Dense) | (None, 1028) | 2632708 |
| dense_3 (Dense) | (None, 70,374) | 72414846 |
| Total params: 112,392,834 | | |
| Trainable params: 112,392,834 | | |
| Non- trainable params: 0 | | |

We tested the proposed network using both the conventional CII-based 3D sectional images (see Figures 4 and 5) and the photon-counted 3D sectional images (PCII), see Figures 6 and 7. Figure 4 shows the reconstructed 3D sectional images using conventional

integral imaging at various distances. It is evident from Figure 4 that the reconstructed sectional images contain both the focused and off-focused points. Figure 5 depicts the output images (after passing through the DL model) containing only the focused-only points at the corresponding depth locations. Similarly, Figure 6 depicts the photon-counted sectional images that are simulated, as explained in Section 2, at the same depth locations of CII in Figure 4. The corresponding focused-only PCII depth images (i.e., after passing through our proposed DNN) are given in Figure 7. It is evident from Figure 7 that the removal of off-focused points from the reconstructed 3D sectional images (both in CII and PCII cases) enhances the visual quality of a reconstructed 3D scene would be advantageous for high-level image analysis such as 3D object tracking, segmentation, classification, and recognition, etc.

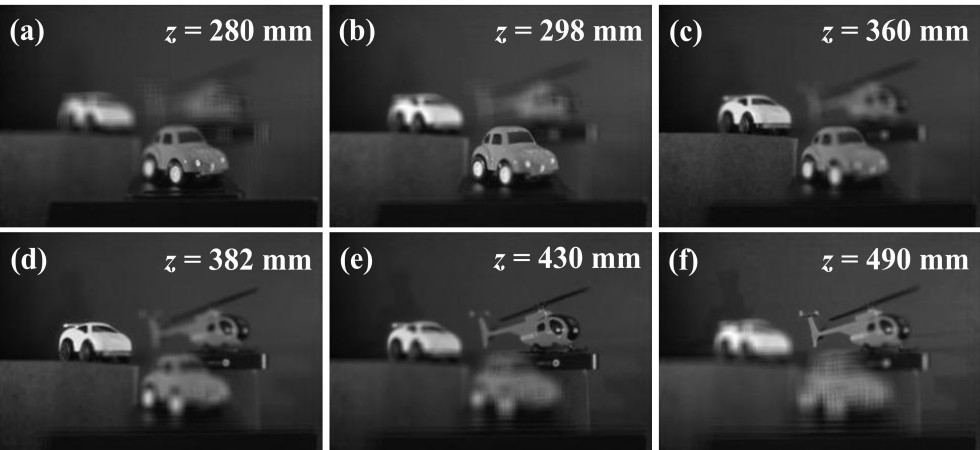

**Figure 4.** Reconstructed 3D CII sectional images at various depth locations.

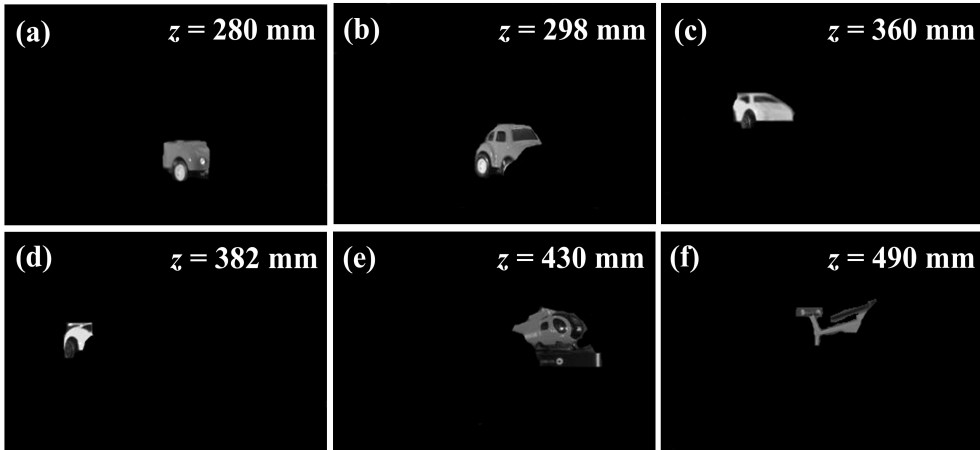

**Figure 5.** Reconstructed focused-only CII sectional images by using the proposed DL network.

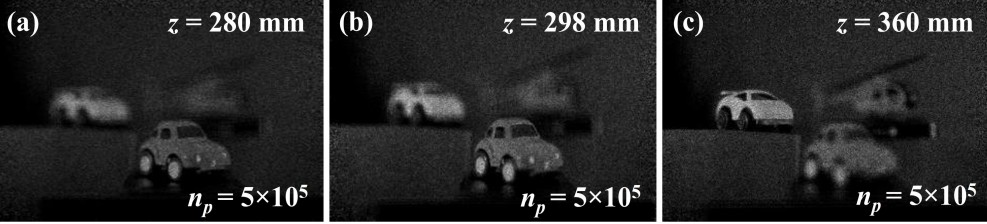

**Figure 6.** *Cont*.

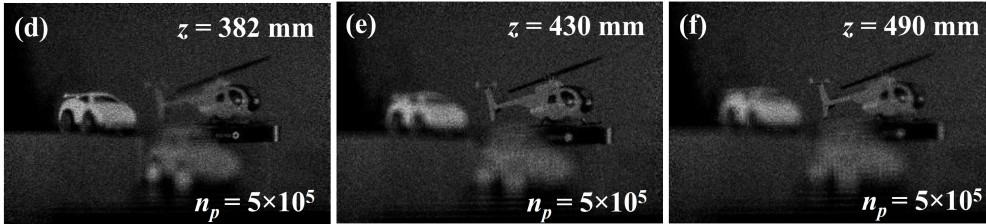

**Figure 6.** Reconstructed 3D PCII sectional images at the same depth locations as of CII. Number of photons ($n_p$) per depth image is $5 \times 10^5$.

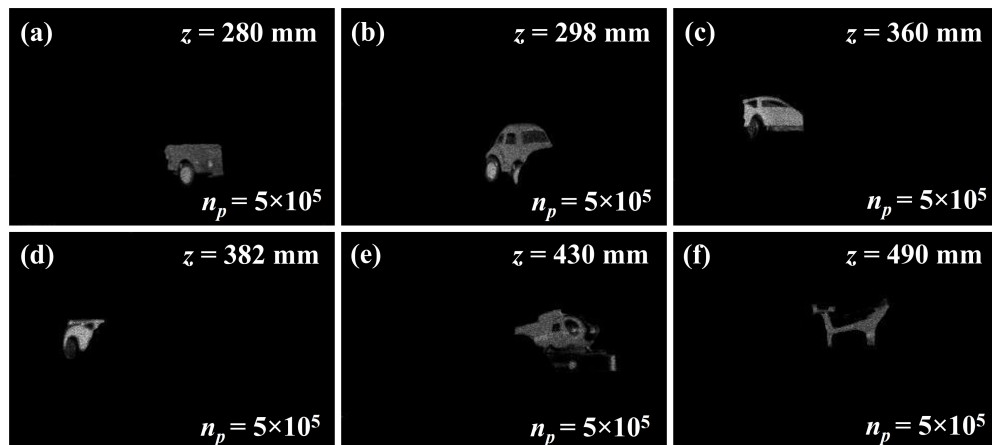

**Figure 7.** Reconstructed focused-only PCII sectional images using the proposed DL network. Number of photons ($n_p$) per depth image is $5 \times 10^5$.

## 4. Conclusions

In this paper, a method that automatically discards the off-focused voxels from the conventional computational integral imaging (CII) and the photon counted 3D sectional integral imaging (PCII) is proposed. To achieve this, we developed a six-individually ensembled supervised deep learning network (i.e., dense neural network) that efficiently removes the off-focused points while simultaneously reconstructing the focused-only points. The proposed network takes the 3D sectional images that contain both the off-focused and the focused portions (pixels). For data pre-processing, we used the Otsu thresholding technique to remove the unwanted background. These processed images are then used to train our proposed network. The trained model is tested against both the conventional CII and the maximum likelihood-based photon-counted 3D sectional images. We believe the removal of off-focused points from the 3D sectional images aids with high-level image analysis such as particle detection and tracking.

**Author Contributions:** I.M. planned the project; S.P. and V.C.D. performed simulations; J.T.S. mentored the project. All authors have read and agreed to the published version of the manuscript.

**Funding:** This research was funded by the Department of Science and Technology (DST) under the Science and Engineering Research Board (SERB) grant number SRG/2021/001464.

**Institutional Review Board Statement:** Not applicable.

**Informed Consent Statement:** Not applicable.

**Data Availability Statement:** Data for this paper is not publicly available but shall be provided upon reasonable request.

**Acknowledgments:** V.C.D. acknowledges the support of the SRM University AP research fund. S.P. and I.M. acknowledge the Science and Engineering Research Board (SERB) under SRG/2021/001464. I.M. thank Bahram Javidi of the University of Connecticut and Inkyu Moon of DGIST for providing the dataset. I.M. eternally thank the late Prof John T Sheridan for all his support. Correspondence and requests should be addressed to Inbarasan Muniraj (inbarasan.muniraj@alliance.edu.in).

**Conflicts of Interest:** The authors declare no conflict of interest.

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
