# Peer review of "A Deep Learning Framework to Remove the Off-Focused Voxels from the 3D Photons Starved Depth Images"

_photonics, doi:10.3390/photonics10050583_

Round 1

Reviewer 1 Report

This paper proposes a DNN model that comprises six different DNN models, As an ensembled supervised deep learning method that efficiently removes the off-focused points while simultaneously reconstructing the focused-only points. It has some technical innovation and application value, which can provide researchers with ideas and information. However, there are some problems with this article and it is better to revise it before publishing it.

1. The section on Sx and Sy in line 86 of Part2.1 is best listed and explained as a separate formula rather than mixed in with the explanation of formula (2). In the meantime, it's better to explain what V and T mean.

2. Formula (3) in Part 2.2 and Wn in interpretation should adopt the same form of upper and lower scripts.

3. Numerical comparison between this method and traditional PCII method at the end of 3.Experimental Results will help to enhance persuasiveness.

4. In Conclusion, "Initial study shown that a single DNN model trained with a complete dataset was unable to form a proper input-output relationship. "is not clearly reflected in the text, and there is no presentation of single DNN model processing results or conclusions cited in other literature.

5. In "Abstract:" part, the research background should be simplified as far as possible, and the focus should be placed on technological innovation.

6. "Sensing in Dark" in "Keywords" is not very precise.

7. It is recommended to follow the recent papers to increase the visibility of this paper. https://doi.org/10.1016/j.mtcomm.2022.103375 https://doi.org/10.1016/j.nanoen.2022.107819

Reviewer 2 Report

This paper proposes a novel method for an ensembled Deep Neural Network (DNN) to automatically identify and remove the off-focused voxels from the 3D scenes that are reconstructed using both the conventional CII and PCII techniques. Results demonstrate that the proposed DNN can efficiently discard the off-focused points and reconstruct a focused-only 3D scene with an accuracy of 98.57%. The manuscript is well-written, and the analytical details are well-documented. I think the authors should add a discussion of computation time to enrich the article, I believe it is acceptable for publication.

Reviewer 3 Report

General Comments:

The paper deals with the reconstruction of 3D scenes under photons-starved illumination and presents a deep-learning approach to improve this reconstruction via discarding the redundant off-focused voxels. The topic is important in 3D image processing and the presented approach could improve some applications like 3D object tracking and biomedical 3D imaging. However, the presentation lacks sufficient reasoning and details.

Specific Comments:

1.    It is unclear how the DNN can discard the off-focused voxels. Is there any link to Equations (1) and (2) and their parameters? A clear presentation is necessary.

2.    The input to the DNN in Equation (3) is unclear. Is it a 2D image data? How are the 6 DNNs integrated? Details are necessary.

3.    The training dataset and the trading process are unclear.

4.    There should be a clear performance measure to be used for comparing the results with existing approaches.

5.    Please justify the number of the nodes and the number of hidden layers that are required for optimal performance. These numbers would affect the complexity of the proposed method.

Round 2

Reviewer 3 Report

The Authors have addressed all of the Reviewer’s comments sufficiently.

The revised version is useful and suitable for publication in MDPI Photonics.

Author Response

Many thanks for giving us a chance to improve our manuscript. The comment regarding total computational time has been added in the revised manuscript. Please find the same in the attachment.